# Monitoring of the Forgotten Immune System during Critical Illness—A Narrative Review

**DOI:** 10.3390/medicina59010061

**Published:** 2022-12-28

**Authors:** Maria A. Serrano, André M. C. Gomes, Susana M. Fernandes

**Affiliations:** 1Serviço de Medicina Intensiva, CHULN, 1649-028 Lisboa, Portugal; 2Instituto de Medicina Molecular, FMUL, 1649-028 Lisboa, Portugal; 3Clínica Universitária de Medicina Intensiva, FMUL, 1649-028 Lisboa, Portugal; 4Grupo de Investigação e Desenvolvimento em Infeção e Sépsis, 4450-681 Matosinhos, Portugal

**Keywords:** flow cytometry, immune system, monitoring, critical illness

## Abstract

Immune organ failure is frequent in critical illness independent of its cause and has been acknowledged for a long time. Most patients admitted to the ICU, whether featuring infection, trauma, or other tissue injury, have high levels of alarmins expression in tissues or systemically which then activate innate and adaptive responses. Although necessary, this response is frequently maladaptive and leads to organ dysfunction. In addition, the counter-response aiming to restore homeostasis and repair injury can also be detrimental and contribute to persistent chronic illness. Despite intensive research on this topic in the last 40 years, the immune system is not routinely monitored in critical care units. In this narrative review we will first discuss the inflammatory response after acute illness and the players of maladaptive response, focusing on neutrophils, monocytes, and T cells. We will then go through commonly used biomarkers, like C-reactive protein, procalcitonin and pancreatic stone protein (PSP) and what they monitor. Next, we will discuss the strengths and limitations of flow cytometry and related techniques as an essential tool for more in-depth immune monitoring and end with a presentation of the most promising cell associated markers, namely HLA-DR expression on monocytes, neutrophil expression of CD64 and PD-1 expression on T cells. In sum, immune monitoring critically ill patients is a forgotten and missing piece in the monitoring capacity of intensive care units. New technology, including bed-side equipment and in deep cell phenotyping using emerging multiplexing techniques will likely allow the definition of endotypes and a more personalized care in the future.

## 1. Introduction

Critical illness is defined as the presence of organ dysfunction following an acute insult, requiring medical interventions to restore homeostasis. Immune dysfunction is common in critical illness, not only in the setting of infection, but also after trauma, single organ infarction and neurocritical illness, among others [1].

The immune organ, in a simplified description, is a system capable of interacting with the self and the external environment, distinguishing between the two and hence guaranteeing a defense from external agents. It consists of an innate arm that allows a quick response to endogenous and exogenous pathogenic stimuli, and an adaptive arm that, upon antigenic recognition, produces a pathogen-specific immune response, developing memory of the recognized antigens. Communication between these two arms takes place through direct contact between cells, as well as through free molecules with an autocrine, paracrine, or systemic effect, which includes cytokines and chemokines, among others [2]. The function of the immune system is, however, not limited to pathogen response but has a fundamental role in tissue regeneration processes, both frequently disturbed during critical illness.

Notwithstanding, in-depth monitoring of the immune system in critically ill patients has never become common practice, despite both broad research in the subject for the last 40 years and common use of immune modulators like corticosteroids [3]. Despite, rudimental monitoring of the immune response to track response to antibiotic therapy is already in used, immune monitoring could be further enhanced to guide modulation strategies, like corticosteroid therapy widely used in multiple critical ill scenarios or the administration of monoclonal antibody therapy, such as tocilizumab (anti-IL-6 receptor) recently used during the COVID-19 pandemic [4]. In sum, multilevel immune monitoring is an important step towards the much-needed personalization of therapeutic interventions (Figure 1) [5].

In this review, we will briefly introduce the acute inflammatory response in critical illness, discuss why monitoring is needed and list some of the most promising markers based on flow cytometry techniques.

## 2. Why Monitor the Immune System during Critical Illness?

Tissue and organ lesion and injury are the hallmark of patients admitted to the ICU. Critical care physicians are familiar with brain, heart, lung, gastrointestinal or kidney monitoring, but not with immune monitoring, nor how it reflects adequate or inadequate immunological function. 

Alarmins released from injured and dying cells (DAMPs -damage-associated molecular patterns) together with molecules originated from microbes (PAMPs—pathogen-associated molecular patterns) determine immune cell activation [6]. Responding immune, epithelial, or endothelial cells express alarmin receptors (PPR—pattern recognition receptors) which include TLR (toll-like receptors), C-type lectin receptors, nucleotide-binding oligomerization domain-like receptors, retinoic-acid-inducible gene-I-like receptors and RAGE [6,7]. One of the most studied is TLR4, that recognizes LPS molecule. Genetic polymorphisms associated with this molecule have been associated with an increased risk of critical illness such as sepsis or multiple organ failure after trauma [8,9]. Aside from TLR4, many other PPR have been implicated in the pathophysiology of critical illness (Figure 2) and strategies to modulate them might, in the future, help reprogram the immune response in the context of critical illness.

The downstream intracellular signaling of DAMPs or PAMPs involves different proteins, a major one being MyD88, resulting in the subsequent activation of NF-kB and the transcription of early cytokine genes, such as *IL-1B, IL-8* or *IL-6,* and interferon genes [6].

Neutrophils are the most prevalent cells in circulation and constitute the first effectors of the innate inflammatory response. Upon activation, neutrophils express chemokine receptors such as CXCR2, which recognize molecules released by endothelial cells attracting neutrophils into the interstitial space [10]. Furthermore, neutrophils can rapidly be recruited by release from capillary beds, such as in the spleen and lung, where they have a slower transit time [11,12].

Neutrophils are armed with potent antimicrobial molecules that, when released freely like in degranulation or formation of extracellular webs, can also cause tissue damage. The latter response, whilst being fundamental for neutrophil function, is at the base of the pathophysiological process of pathologies such as ARDS [13] or ischemia-reperfusion injury [14].

Simultaneously, once activated, dendritic cells migrate to the lymph nodes, and present antigen to CD4 T lymphocytes, recruiting the adaptive immune response. These later produce cytokines, like IL-8 or IL-17, that further recruit neutrophils, thus amplifying the immune response. In addition to dendritic cells, monocytes change their phenotype from M2 (anti-inflammatory) to M1 (pro-inflammatory), produce pro-inflammatory cytokines and present antigens thus contributing to set off the adaptive immune response.

This response consists of antibody production, depending on B cell maturation and differentiation into plasma cell. T cell associated responses involve either cytotoxic (CD8) T cells, particularly relevant in viral infections, or T-helper (CD4) cells, fundamental to cytokine production and for shaping myeloid and B cell responses. Recruitment of each cell type depends on their upregulation of chemokine receptors, which also reflects the cell’s function, and the local tissue production of chemokines. This allows the infiltration of different cells in tissues vital for the response against foreign agents or necrotic tissue.

In parallel to the above, tissue regeneration is promoted. This supports a shift to regulatory mechanisms, with an increase in regulatory T cells (Tregs) in tissues, in Th2 mediated response [15], and the production of cytokines such as TGFβ. Tissue regeneration and complete recovery depends on stopping the inflammatory process. Dysregulation at this stage, enabling chronic inflammation, may contribute to the persistence of organ dysfunction and accumulation of fibrotic tissue [16].

Immune dysregulation can occur at several moments of critical illness combining immune activation and suppression mechanisms. Cytokine storm with consequent multiple organ failure is considered a pathogenic hallmark in distributive shock associated with sepsis, trauma, or ischemia-reperfusion injury. In this context, high levels of cytokines such as IL-1, IL-6, IL-18, IL-8 or TNF contribute to organ lesion by recruiting neutrophils, NK cells and other lymphocytes to the tissues, by activating the endothelium and overall inducing a new metabolic immune state, as recently reviewed in the context of COVID-19 [17].

After this acute phase, there is either complete recovery, death, or progression towards an occult persistent immune dysregulation. This later phenomenon can occur in all settings of critical illness [18], but it was in sepsis where it was primarily described. Immune dysregulation has been associated with an increased risk of nosocomial infections [19], as well as cognitive dysfunction, ICU acquired muscle weakness and increased mortality [20]. It is clinically represented by a group of patients with prolonged critical illness (more than 14 days of hospitalization in the ICU, persistence of organ dysfunction with evolution to chronicity in some cases) [21]. Patients with this immune dysfunction are also represented in cohorts of patients with post-intensive care syndrome (PICS), as defined in 2012 by Moore, and immune dysfunction may partially explain its development [22].

The persistent immune dysregulation is characterized by a persistent activation of the innate system, by DAMPS or PAMPS. If the initial stimulus of the septic episode starts through PAMPS such as LPS, immune activation mediated by PAMPS might persist due to reactivation of viruses such as CMV, EBV, HHV-6, or TTV, occurring in more than 40% of septic patients [23], or due to nosocomial bacterial infection. At the same time, there is the release of DAMPs such as S100, nuclear or mitochondrial DNA, HMGB1, RAGE, IL-33, adenosine, amongst others, that perpetuate the inflammatory cascade in response to injured tissue [24].

This persistence of alarmins causes an exhaustion of the immune system, which is expressed by the absence in lymphopenia recovery, a decrease in the expression of human leukocyte antigen DR (HLA-DR) in monocytes, and an increase in sPD-L1, which might support a state of cellular anergy and a decreased response to new pathogens. This immune-suppressive state can be seen as an adequate adaptive response to persistent cellular activation and an attempt to enter the repair process. Whatever the interpretation, this immunological picture has been called “immune paralysis” and is characterized by (a) a lower ability to present antigen, (b) a lower production of inflammatory cytokines and (c) a reduced clearance of bacteria. This phenomenon determines a greater risk of nosocomial infections and a persistent chronic inflammatory state. It also contributes to ongoing muscle catabolism marked by high levels of GLP-1 (15), whose pathophysiology is related to mitochondrial damage and release of mitochondrial DNA and molecules derived from reactive oxygen species. This process of prolonged immune dysregulation is also associated with accumulation of myeloid suppressor cells, which can be monocytic (M-MDSC) or polymorphonuclear (PMN-MDSC). These cells can also suppress lymphocyte function and decrease cytokine production, contributing to decreased pathogen clearance. The increase in myeloid suppressor cells is associated with a greater infection risk, ICU length of stay, and greater mortality [25,26].

In addition to changes described in the innate system, multiple alterations in lymphocytes are also well-known. These range from persistent lymphopenia to modification in the profile of CD4 and CD8 T lymphocytes as well as of B cells [27], promoting dysregulated immunoglobulin [28], interferon and cytokine production and clearance, all contributing to organ dysfunction.

The presence of immune dysregulation is frequently observed in chronically ill patients. Nevertheless, even this group is heterogeneous and might demand different strategies. So, carefully assessing the immune system and monitoring throughout chronic disease might have some benefits. For instance, we could better guarantee the enrolment of more homogeneous groups of patients in clinical trials and in the future cater personalized medicine. Although it is an area of active research, the clinical application of specific biomarkers to aid monitoring the use of therapeutics in intensive care medicine such as corticosteroids, or the risk assessment of secondary complications such as hospital-acquired infection and immunosuppression associated with critical illness, is yet to be seen [5].

In this field, data reanalysis of several randomized clinical trials have allowed to recategorized patients into various phenotypes and reaccess the intervention results. In ARDS, for example, patients were classified into a pro- and anti-inflammatory phenotype profile, based on multiple parameters of which the most immunologically relevant were IL-6, IL-8, TNFr1 and ICAM-1 [29]. This stratification allowed us to understand, for example, that the effect of fluid restriction strategies [30] was not transversal between phenotypes and could even be deleterious in the hyper-inflammatory group [31].

From these studies also emerges the concept that only a cluster of markers and not isolated markers will allow the definition of homogeneous phenotypes. Artificial intelligence systems facilitating the identification of clusters from clinical and lab data, associated with an increased technical capacity at the bed side to simultaneously measure multiple molecules, may aid us in the future to apply this paradigm [32].

## 3. Old Markers, any New Information?

The monitoring of the inflammatory response is currently done with generic tests that do not allow to distinguish the type of response or the etiology of inflammation. 

### 3.1. Leukogram and Neutrophil/Lymphocyte Ratio

The leukogram, one of the most frequently requested complementary test, is frequently under-interpreted. After an acute inflammatory response, and upon release of adrenaline, neutrophil demargination occurs, decreasing the transit time of these cells in the lung or spleen, thus contributing to their rapid increase in number. This response is instantaneous, brief, and unspecific to infection, yet it is a sensitive marker of an inflammatory or adrenergic response. Instead of neutrophilia, transitory neutropenia is also common, particularly in sepsis, which may result from a decrease in the neutrophil’s average lifespan [33,34].

In addition to absolute and relative neutrophil counts, the predictive value of neutrophil-lymphocyte ratio (NLR) for mortality has been demonstrated in multiple situations and can be easily incorporated into clinical practice. In fact, the neutrophilia is commonly accompanied by an abrupt decrease in the lymphocyte count during critical illness, which, if persistent, is associated with higher mortality [35]. There are potentially multiple reasons for this lymphopenia—increased apoptosis following rapid increase in pro-inflammatory cytokines, massive migration to tissues, decreased lymphopoiesis as an acute response to pathogenic stimuli, however a unifying pathophysiological mechanism has not yet been described. NLR has been associated with increased mortality in SARS-CoV-2 infection [36], abdominal trauma [37], pneumonia [38] and acute pancreatitis [39].

Thus, observing the NLR in critically ill patients (normal range: 1–2; pathological above 3 and below 0.7), can be useful in the diagnosis and follow-up of inflammatory situations [40], signaling clinical improvement or deterioration [41].

### 3.2. Soluble Biomarkers

Soluble markers like C-reactive protein (CRP), procalcitonin, or pancreatic stone protein (PSP) are used at bedside and help to guide clinical decisions, particularly antibiotic treatment initiation and duration (Table 1). However, their intrinsic immune functions and role within the immune response are frequently overlooked.

CRP is mainly produced in the liver, as well as by smooth muscle cells, macrophages, endothelial cells, lymphocytes, and adipocytes, in response to IL-6. So, in any given clinical situation with high IL-6 levels there may also be high circulating levels of CRP. CRP, in its monomeric or pentameric form, binds to complement molecules, contributing to the opsonization of microorganisms, activation of neutrophils and monocytes, and stimulation or inhibition of the inflammatory response, depending on the form in which it is presented (monomeric or pentameric) [65]. Most infectious diseases elicit a common immune response, and so it is unsurprisingly that for diagnostic purposes, CRP’s predictive value is low and it’s use is not recommended when deciding on antibiotic use. In contrast, monitoring CRP may be useful to evaluate the response to interventions [66]. In fact, for patients with community or hospital-acquired pneumonia, a halving of CRP value at 72 h post antibiotic regimen initiation was associated with better prognosis and an effective antibiotic response [42,67,68]. On the other hand, we must highlight that most existing studies have relevant methodological gaps and are very heterogeneous, not allowing for firm conclusions to be drawn in narrative reviews [69] or meta-analysis [70]. 

Procalcitonin is a molecule produced in both the parathyroids and in adipose tissue. In the former, it is secreted as calcitonin, and depends on calcium and vitamin D levels, and in the latter, it is released as procalcitonin in response to inflammatory stimuli such as IL-1 or IL-6 [71]. Its expression in adipose tissue is inhibited by IFNγ (the main cytokine involved in the anti-viral response) and by IL-17, released at higher levels during fungal infections, partially explaining its lower levels in these settings. Its use is also not recommended when diagnosing sepsis (48). Although an elevation in procalcitonin levels is associated with a greater probability of severe infection and bacteremia, it does not have sufficient sensitivity or specificity to exclude or diagnose sepsis. This has been evaluated in multiple infection diagnostic protocols, with disparate results [72,73]. Its use in some specific contexts like respiratory infections, is somehow more consistent [74], but not enough to guide antibiotic treatment or establish prognosis. Procalcitonin should preferably be used as a monitoring molecule for patient response and to promote earlier discontinuation of antibiotic therapy [75,76], particularly in contexts where prolonged periods of antibiotic therapy are anticipated [77,78].

Pancreatic stone protein (PSP) is an acute-phase protein that binds to neutrophils and determines their activation, which may promote bacterial aggregation [79]. This protein is produced in the pancreas, and its local function is not completely known. After non-pancreatic tissue injury, there is an increase in its production and release into the bloodstream by the pancreas. Like CRP or even PCT, PSP also increases in multiple circumstances, such as trauma [49], however it might be more specific then PCT for diagnosing infection [80]. In a meta-analysis published in 2021, Prazak et al. sought to aggregate all studies that aimed to assess the diagnostic capacity of PSP in critically ill patients with infection concluding that a cut-off of 44.18 ng/mL has greater specificity than either PCR or PCT [80]. Notwithstanding, in patients undergoing cardiothoracic surgery, the predictive value for infection of this marker was lower, with a precision of only 0.76, meaning surgical technique or the use of extracorporeal circulation was not impacting its values [81].

However, none of these soluble markers help define immune dysfunction, and do not reflect the overall host response to a danger stimulus.

In this regard, cytokine and chemokine quantification might be closer to help define immune responses.

The quantification of serum cytokines has been performed in multiple research settings and has changed the comprehension of critical illness pathophysiology. This allowed to create endotypes for multiple critical ill syndromes like ARDS or sepsis [29,82,83]. Increasing availability of point of care tests for the quantification of these molecules, may in the future help integrate them in clinical decision protocols, targeting infection and immune dysfunction diagnosis (Table 1). 

Some of these markers have already been used in clinical practice. IL-6 assays are used to monitor rheumatoid arthritis patients in clinical follow-up or integrated in clinical trial design [84]. In critically ill patients, IL-6 measurements were massively used during the pandemic and helped guide the use of tocilizumab [85,86]. In addition, increased levels of IL-10 (an immune suppressor molecule) and IL-8 (neutrophil recruitment molecule) have popped-up as relevant prognostic molecules in most large database studies [87,88]. 

Nevertheless, none of these biomarkers are surrogates for cell associated immunity, which we will discuss in the next chapter.

## 4. Can We Monitor the Immune System in Depth? The Clinical Application of Flow Cytometry

The frequency and absolute numbers of immune cells, such as monocytes, neutrophils or lymphocytes, or their sub-types, such as CD4 + T cells—further subdivided into TH1, TH2, TH17 and regulatory T cells (Tregs)—can be used to monitor the immune system. Flow cytometry has fueled the identification of these immune subsets and their monitoring in different disease settings [89,90]. Currently, completely automatized flow cytometry methods are used for performing routine full blood counts or lymphocyte phenotyping (T, B and NK cells) commonly used in situations like HIV infection or after treatment with rituximab [91].

Flow cytometry can rapidly analyze multiple immune populations in solution at the single cell level. Cell shape and complexity can be inferred, as well as the presence of a specific cell associated protein when immunoassayed with fluorescently conjugated antibodies [92]. In addition, to cell associated markers, when coupled to cell stimulation assays, flow cytometry-based techniques can be used to determine cell function, such as per cell cytokine production, oxidative function, cytotoxic activity, and to ascertain cell division [93,94]. Besides cellular immunophenotyping, flow cytometry can also be applied to detect and quantify soluble proteins, such as cytokines and chemokines, in multiplexed assays with the equivalent assaying power of 100 Enzyme-Linked Immunosorbent Assays(ELISA) assays [95]. It can also be used to determine an array of inflammatory mediators allowing a better discrimination for diagnosis [96], and, in a recent study, quantitative flow cytometry was used to assess the number, viability and drug-resistance of common disease-causing bacteria [97].

### 4.1. Limitations and Challenges to the Use of Flow Cytometry

There are, however, some downsides to flow cytometry that hinder its use for clinical immune monitoring. Flow cytometry is an open technique, with many different analyzers and often homemade protocols that lead to variation in results and their interpretation. This leads to decreased reliability and major issues in standardization [98] crucial for comparing results between centers. These are now starting to be overcome [99]. Standardized immunostaining protocols between labs, calibration, and daily quality control of flow cytometers with specific beads, the use of same batch antibodies with stable fluorophores or the use of calibrated beads that convert fluorescence intensities to numbers of antibodies bound per cell, are some of the strategies to implement reliable flow cytometry protocols. Besides standardization issues, this technique requires specialized technicians and increasingly complex and expensive instruments which availability can be challenging in some centers [100,101,102]. Nevertheless, the investment in cutting-edge flow cytometry techniques for critical care will surely prove to be fruitful, like it already is for deep immune monitoring in transplant patients, or those receiving immunotherapy for several malignancies [103,104].

### 4.2. New Techniques for in Depth Monitoring

A recent upgrade, spectral flow cytometry, is simplifying the access to in-depth immunophenotyping. As opposed to conventional flow cytometers, spectral cytometers capture the full spectral emission of each fluorophore. Therefore, while it is already challenging to assess 18 markers by conventional analyzers, spectral flow cytometers allow for the relatively easy detection of more than 40 markers per cell. While they require similar controls as with conventional cytometers, spectral analyzers possess universal instrument settings and allow for a standardized output across all instruments, a clear advantage during multi-center clinical studies. Nevertheless, these instruments’ availability is still low, and their cost elevated.

Another method allowing the analysis of over 50 markers is mass cytometry. In this technique, also known as cytometry by time-of-flight (CyTOF), instead of using fluorophores, the antibodies are conjugated with heavy metal reporter ions and cells are analyzed by time-of-flight mass spectrometry to quantify the isotopic masses and, thus, the bound antibodies and the expression of markers of interest. This technology reduces the problems related with spectral overlap and sample autofluorescence present in flow cytometry, and the wide availability of heavy metal isotopes allows for multiplexing. Besides the assessment of surface cell lineage markers, mass spectrometry allows for the simultaneous quantification of many intracellular targets, such as cytokine production, transcription factors and protein phosphorylation, which can inform on cell states and response to stimuli. More recently, this technique was repurposed to allow for the multiplexed imaging of tissue markers, which would permit clinicians to understand immune pathology at tissue level [105,106].

Several other ways to monitor the immune system in-depth are growing more popular nowadays. For instance, gene expression profiling by RNA-sequencing techniques is allowing for the comprehension of bulk or single cell heterogeneity in homeostasis and disease. Moreover, spatial biology methods emerging in the recent years allow for the extraction of spatially resolved molecular information from tissue biopsies.

Therefore, there are a plethora of techniques available that can in the future assist in clinical decisions as well as fuel translational research to improve critical care.

Next, we will discuss some of the more promising cell associated immune markers and how they might shape clinical decisions.

## 5. What Are the More Promising Immune Cell Associated Markers?

The assessment of the inflammatory response is undoubtedly incomplete if it is only based on soluble markers and if it does not consider the cellular phenotype and per cell expression of biomarkers (Table 2). These reflect specific immune alterations and contribute to a better characterization of the immune profile.

### 5.1. T-Cell Associated Markers

As previously stated, acute inflammation is frequently characterized by lymphopenia, and its persistence is associated with worse prognosis [118]. Although the mechanisms are not completely understood, it is likely that IL-7 responses contribute, and phase II IL-7 trials in sepsis are ongoing [119]. The levels of IL-7 receptor can easily be monitored in different immune cells by targeting CD127 (IL7 receptor alpha) [120] and are associated with mortality, particularly if low at day 3 of septic shock [121]. In addition to T cells counts normalization, retaining a diverse T cell repertoire [122] and a balanced frequency of each phenotype is probably relevant for full return to homeostasis. All these parameters can be assayed. For instance, persistent increase in Treg frequency has been linked with increased risk for secondary infection and persistent organ dysfunction, as recently reviewed by Gao and collaborators [123]. Also, high levels of PD-1 expression on T cells, reflecting T cell response to high and persistent activation levels, might impair T cell ability to respond [124]. For the patients with this profile, repurposing immune therapy already used in cancer could help prevent and treat nosocomial infections [125].

### 5.2. Monocyte Associated Markers

Monocytes, which represent about 10–20% of circulating leukocytes, are involved in the process of amplifying the inflammatory response. Although there are essentially two phenotypes described, M1 (inflammatory) and M2 (non-inflammatory), there are intermediate stages and plasticity between them [126]. Monocytes are antigen presenting cells that modulate the adaptive and innate response and influence the type of T cell response. Antigen presentation is dependent on the number HLA-DR molecules. Its expression in monocytes in patients with sepsis has been exhaustively studied and protocols for flow cytometry standardization have already become available [127]. More importantly, the trajectory of HLA-DR levels stratifies patients [128] and it has been used in small clinical series to guide immune stimulation with Interferon γ [129]. This was also measured in COVID-19 cohorts with similar results [130]. Strikingly, in some patients the levels of this molecule only return to normal values 6 months post illness [131].

Persistent decrease in HLA-DR in monocytes (CD14 + cells) is associated with higher mortality with significant differences for survivors at days three to four of the septic episode. Four sepsis-response endotypes can be described [128]. For patients whose levels do not increase or who have a progressive decrease in monocyte associated HLA-DR levels, the prognosis is worse. In addition to prognostic classification, the level of HLA-DR in monocytes could be used to assess responses to immunomodulatory therapy such as IFNγ in patients with infection or persistent lung injury [132].

### 5.3. Neutrophil Associated Markers

Another relevant molecule extensively studied in critically ill patients is CD64, an FC gamma receptor, which binds to the FC portion of antibodies, and is constitutively expressed on monocytes and at very low levels on non-active neutrophils. The expression of this receptor on neutrophils increases in response to inflammatory cytokines produced in the presence of external agents such as bacteria or after exposure to endotoxin [133]. As for HLA-DR, the quantification and use in clinical practice of this marker depends on standardized flow cytometry techniques. In a prospective observational study, its potential for monitoring critically ill patients was evaluated, including 468 patients, of which 103 had sepsis. With a cut-off of 230 in mean fluorescence intensity (MFI), and particularly if combined with abnormal CRP values, a probability for sepsis of 92% was described; importantly, if values were both normal, sepsis could be excluded with 99% confidence [134]. In 2015, a first meta-analysis was published that included 8 studies and 1986 patients. Since the cut-off used in each study differed, it was difficult to combine results. Nevertheless, an AUC-ROC value of 95% was assigned for the diagnosis of infection using the cut-off of 230, with a particularly high specificity [135].

### 5.4. Combination of Markers

At the end, most of the studies addressing immune function use combination of biomarkers. For example, in the multicentric English INFECT study, authors evaluated CD88 levels on neutrophils, HLA-DR levels on monocytes and the frequency of Tregs. All were associated with an increase increased risk of secondary infections (from 2.18 to 3.44 increase in odds ratio) particularly from day 3 through 9 [136].

Nowadays, in critical care units, more than one variable is used for stablishing a hemodynamic profile, likewise more than one marker can be expected to be needed to create reliable endotypes that reflect immune organ function.

## 6. Future Perspectives and Challenges

Critical care syndromes will be progressively defined, and new clinical entities will be identified. Some of them will be based on the immune response to a bug or a danger stimulus [137]. In-depth monitoring the function of the immune system will then be necessary to individualize treatments according regulated pathways. This will contribute to personalize care in the ICU. Nevertheless, for immune monitoring in this field to become tangible, there is still a large knowledge gap to be filled in immune response trajectories and the markers that better define them. Even for the most promising markers like HLA-DR expression by monocytes, clinical trials are still needed that associate the use of this knowledge with patient centered outcomes. Importantly, despite the overall goal to scientifically prove the benefit of immune monitoring, new trial designs must be implemented to overcome sample size and inclusion criteria must consider patients more likely to benefit from this individualized approach. For instance, including all septic shock or ARDS patients will lead to negative studies. A way to go could be to use this strategy only in highly complex patients, such as the ones fulfilling chronic critical illness criteria, or the ones in whom corticosteroid treatment is considered.

In addition, immune monitoring is likely to increase costs in the management of ICU patients, and cheaper tools must emerge to allow the application of tools like flow cytometry.

Finally, specific management strategies according to immune profiles are still to be defined, like immune stimulation strategies, immune based treatments to promote regeneration or specific infection prevention programs. Repurposing the use of already used drugs might be the smarter solution [138].

## 7. Conclusions

Although monitoring the inflammatory response using flow cytometry has not yet been demonstrated to have a prognostic impact, we do believe it will be essential to personalize treatment in critically ill patients. This strategy based on flow cytometry will allow for smarter enrollment in randomized clinical trials, targeting specific populations with immune modulators. It will also help to identify at risk patients for chronic disability and ensure personalized strategies targeting a specific risk profile. At the end, the correct evaluation of the response to therapies, such as antibiotics, corticosteroids, or IFNγ, will help to move critical care in the direction of personalized care.

## Figures and Tables

**Figure 1 medicina-59-00061-f001:**
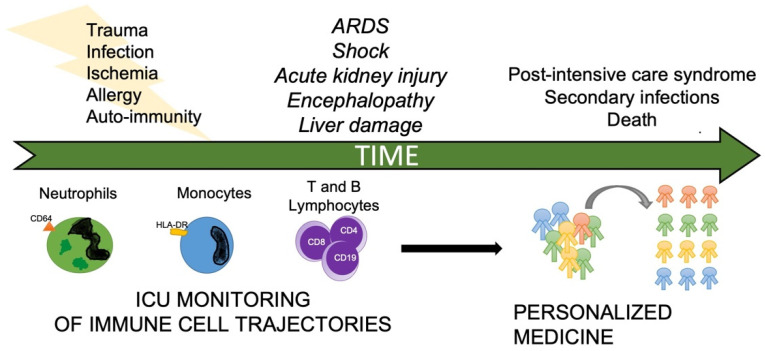
Perspective on the application of immune monitoring in the intensive care unit.

**Figure 2 medicina-59-00061-f002:**
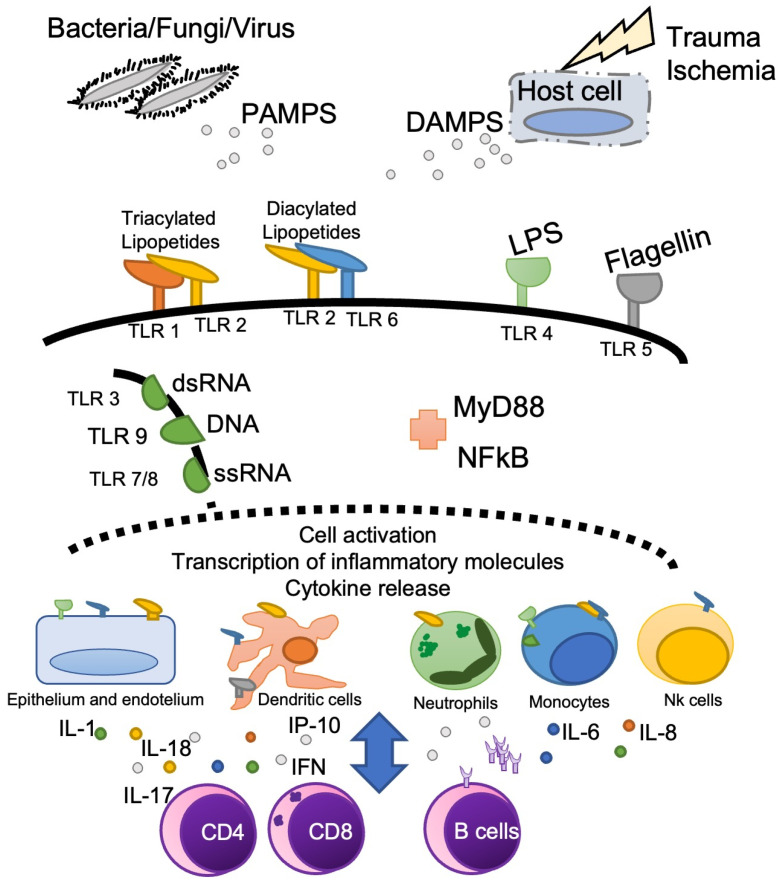
Schematic of initial triggering of inflammation. Main cell populations involved in the process are depicted. DAMPS—damage-associated molecular patterns, ds—double stranded, IFN—interferon, IL—Interleukin, IP—Interferon-gamma induced protein; LPS—Lipopolysaccharide, PAMPS—pathogen-associated molecular patterns, ss—single stranded, TLR—Toll like receptor.

**Table 1 medicina-59-00061-t001:** Serum immune markers measurable in the ICU.

Biomarker	Function	Relation with Outcome	References
C-reactive protein	Bacteria opsonization	High levels not associated with prognosis Decrease in the first 72 h of infection associated with prognosis	Refs. [42,43,44]
Procalcitonin	Immune function unknown	Increased in sepsis and in ischemia High levels associated with mortality in sepsis and pneumonia	Refs. [45,46,47]
Pancreatic stone protein	Immune function unknown	Increased in sepsis, particularly bacterial sepsis Increased also in pancreatitis and trauma	Refs. [48,49,50]
suPAR	Cell adhesion and migration Immune activation	Increased in sepsis and associated with prognosis	Ref. [51]
IL-6	Pro-inflammatory	Elevated in sepsis and trauma Associated with mortality in COVID-19 Associated with developing multiple organ dysfunction in trauma	Refs. [52,53]
IL-8	Neutrophil recruitment, endothelial activation	Associated with mortality in sepsis Prediction of AKI after trauma	Refs. [54,55]
IL-10	Anti-inflammatory	Elevated in sepsis Higher levels in sepsis than in SIRS Elevation associated with prognosis in sepsis	Refs. [56,57]
IL-18	Neutrophil recruitment	Associated with prognosis in ARDS and sepsis	Ref. [58]
IL-1RA	Antagonist for IL-1	Elevated in sepsis and trauma	Ref. [56]
IP-10 (CXCL10)	Chemoattractant for CXCR3 + cells	Associated with severity and prognosis in COVID-19	Ref. [59]
Interferon γ	Antiviral and antibacterial response	Associated with more severe outcome in sepsis Low levels in the chronic critical phase associated with higher infection risk	Ref. [60]
Interferon α	Antiviral and antibacterial response	Associated with severity and ARDS progression in COVID-19	Ref. [61]
IL-17	Neutrophil recruitment	Elevated in sepsis and linked with prognosis	Ref. [62]
IL-33	Promotes shift toward type II immunity	Associated with prognosis in critically ill Low levels linked with hepatic dysfunction	Ref. [63]
sTNFr1	Receptor for TNF	Elevated in the inflammatory phenotype of ARDS Associated with prognosis in ARDS	Refs. [54,64]

ARDS—Acute respiratory distress syndrome, AKI—Acute kidney injury, SIRS—Systemic inflammatory response syndrome, CXCR3—C-X-C Motif Chemokine Receptor 3, TNF—tumor necrosis factor.

**Table 2 medicina-59-00061-t002:** Cell associated immune markers associated with prognosis in critically ill patients.

Cell	Marker	Immune Function	Cohorts Studied	Possible Clinical Use	References
Monocytes	HLA-DR	Antigen presentation	Septic shock Trauma Major Surgery Decompensated cirrhosis	Trajectories identify patients with increased infectious risk Persistent decreased levels associated with prognosis	Refs. [107,108,109,110,111]
Neutrophils	CD64	Neutrophil activation and phagocytic activity	Sepsis	High levels associated with infectious inflammation Associated with mortality in sepsis	Refs. [112,113]
	CXCR2	Neutrophil migration	Sepsis	Higher in infected patients with sepsis	Ref. [10]
T cells	CD8: PD-1	Decrease cell activation	Sepsis COVID-19	High levels associated with mortality in sepsis and increased risk of secondary infection	Ref. [114]
	CD4: PD-1	Decrease cell activation	Sepsis COVID-19	High levels associated with mortality in sepsis and increased risk of secondary infection	Refs. [115,116]
	CD4: CD127	IL-7 receptor	Sepsis	Low levels associated with prognosis	Ref. [117]

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
