# Peer review of "Monitoring of the Forgotten Immune System during Critical Illness—A Narrative Review"

_medicina, 2022, doi:10.3390/medicina59010061_

Round 1
Reviewer 1 Report
The manuscript entitled "The monitoring of the forgotten immune system – a narrative review". Title, abstract and overall rationale of work to some extent is good. However, there are still some major concerns, which needs to be addressed and needs substantial revision.
1) Introduction section: Author explain very well in this section but author need to provide one figure here to explain all important immune response, cell behavior, chemokines and cytokines roles during acute infection. Here it is important to represent all these as a pictorial view.
2) Section (Why monitor the immune system during critical illness?): a) This is the review article and author talking about the PAMPs, DAMPs, PPR and other pathway and important of these during injuries. However, author did not explains details about these all pathways role during injuries and other circumstances. Furthermore, author need to explain about the mechanism of action of theses pathway, cytokines, T cells and others.
b) In this section author only talking about TLR-4 but we know that there are several TLR and they have important role. Author see this line [The MyD88 signaling cascade is essential for TLR2, 4, 5, 7, 8, and 9 (Burns et al., 2003). TIRAP activation is MyD88-dependent and is associated with TLR2 and 4 (Mansell et al., 2004; Bernard & O'Neill, 2013)]. Author be explain details about these and also incorporate one or two diagram in this section.
c) In this section author explain superficial and I suggest to author they need to explain details to understand complete role of all these. However, author did not explain about the most important cells such as Dendritic cell, natural killer cells and many more. Moreover, there are lots of cytokines and chemokines are activated during infection/recover, so here author need to explain and highlight most important cytokines and chemokines role and there pathways.
d) What about the role of humoral immune response and their role in tissue injury please explain.
3) In this section (Old markers, any new information?) author need to incorporate the figure to show the clear picture
4) In this section (Can we deep monitor the immune system using flow cytometry?) author talking about the uses of flow cytometry and their benefit. Of course it is beneficial but the cost of flow cytometry and antibodies are too high and it is not feasible to use all hospital. Author also need to write the limitation of flow cytometry.
5) Author also need to write future prospective of this review.
Reviewer 2 Report
The authors present a summary on the role of Immune Organ failure and alteration of the immune system during critical illness. They also described the possible role of multilevel immune monitoring for personalisation of therapeutic interventions during critical illness and possible biomarker and techniques to be used to monitor immune system dysfunctions.
The topic is interestingly, however, overall, the manuscript does not flow properly.
- The descriptions of Immune Organ failure during critical illness in the introductory section is short and do not flow logically. This makes difficult to clearly understand the point the author is trying to make.
- The chapter dedicate to the flow cytometry should be more general and include state of art techniques that can be used to monitor immune system.
-The conclusions are short and do not analyse all the points underlined in the manuscript; additionally, in my opinion, seems completely focused on the role of flow cytometry to study immune dysfunctions. A better description on the relevance of monitoring the immune system for personalised medicine during critical illness is necessary here.
- Please provide a table that summarises all the marker described in the manuscript. This will improve the reading and understanding of the role of Immune Organ failure and alteration of the immune system during critical illness.
- Please include one or two Figures illustrating the most relevant events associated to critical illness and related immune dysfunctions.
- Typo and grammar errors should be checked.
Round 2
Reviewer 1 Report
I have completed my evaluation of your manuscript and I found authors have addressed all the concerns raised in the previous version of the manuscript and the quality has improved after incorporating required modifications. Therefore, the manuscript may be considered for publication in this Journal. There are some minor issue need to fulfill before publication.
1) Still there are some typographical errors present in the manuscript for example line 49 (s, among others.[2]
2) Figure 1 author must be improve the quality of figure and resolution.
Reviewer 2 Report
The authors have addressed the comments and suggestions made in the first review. The quality of the article has improved.
Minor comment: Please improve quality resolution of Figure 1.
